# Shared and specific signatures of locomotor ataxia in mutant mice

**Ana S Machado, Hugo G Marques, Diogo F Duarte, Dana M Darmohray, Megan R Carey\***

Champalimaud Neuroscience Program, Champalimaud Center for the Unknown, Lisbon, Portugal

**Abstract** Several spontaneous mouse mutants with deficits in motor coordination and associated cerebellar neuropathology have been described. Intriguingly, both visible gait alterations and neuroanatomical abnormalities throughout the brain differ across mutants. We previously used the LocoMouse system to quantify specific deficits in locomotor coordination in mildly ataxic *Purkinje cell degeneration* mice (*pcd;* Machado et al., 2015). Here, we analyze the locomotor behavior of severely ataxic *reeler* mutants and compare and contrast it with that of *pcd*. Despite clearly visible gait differences, direct comparison of locomotor kinematics and linear discriminant analysis reveal a surprisingly similar pattern of impairments in multijoint, interlimb, and whole-body coordination in the two mutants. These findings capture both shared and specific signatures of gait ataxia and provide a quantitative foundation for mapping specific locomotor impairments onto distinct neuropathologies in mice.

## Introduction

Visibly ataxic mouse mutants exhibit varying patterns of neuropathology throughout the brain (*Cendelin, 2014*; *Fortier et al., 1987*; *Goldowitz et al., 1997*; *Lalonde and Strazielle, 2007*; *Lalonde and Strazielle, 2019*; *Mullen et al., 1976*; *Walter et al., 2006*). Although their motor coordination deficits are generally attributed to abnormal cell patterning within the cerebellum (*Arshavsky et al., 1983*; *Orlovsky et al., 1999*), these lines have distinct patterning defects within the cerebellum, varying degrees of extracerebellar involvement, and differences in age of onset (*Cendelin, 2014*; *Lalonde and Strazielle, 2019*). The nature of the motor deficits exhibited by these mice also varies, and can often be distinguished by trained observers (*Berman, 2018*; *Brooks and Dunnett, 2009*; *Hoogland et al., 2015*; *Lalonde and Strazielle, 2019*; *Schiffmann et al., 1999*; *Stroobants et al., 2013*; *Vinueza Veloz et al., 2015*). However, analysis of motor coordination is often limited to low dimensional descriptions of limited specificity that fail to distinguish between related behavioral phenotypes (*Brooks and Dunnett, 2009*; *Lalonde and Strazielle, 2019*). Analysis of locomotor kinematics can provide higher dimensional readouts of locomotor behavior (*Cendelín et al., 2010*; *Gabriel et al., 2009*; *Zörner et al., 2010*), but can still suffer from a lack of specificity due to an abundance of highly correlated measures that ultimately reflect non-specific features such as changes in walking speed or body size (*Batka et al., 2014*; *Cendelín et al., 2010*; *Machado et al., 2015*). A quantitative understanding of the specific nature of gait ataxia in mutants with well-described abnormalities in circuit architecture could provide important clues into neural mechanisms of motor coordination (*Anderson and Perona, 2014*; *Bastian et al., 1996*; *Berman, 2018*; *Brown and de Bivort, 2018*; *Darmohray et al., 2019*; *Datta et al., 2019*; *Kiehn, 2016*; *Morton and Bastian, 2007*; *Powell et al., 2015*; *Sarnaik and Raman, 2018*; *Udo et al., 1980*).

We previously used the LocoMouse system (*Machado et al., 2015*) to analyze the locomotor coordination of mildly ataxic *Purkinje cell degeneration* (*pcd*) mice, in which neural degeneration, particularly early in postnatal development, is largely restricted to cerebellar Purkinje cells,

\*For correspondence:
megan.carey@neuro.
fchampalimaud.org

effectively disconnecting the output of the cerebellar cortex (*Chen et al., 1996*; *Fernandez-Gonzalez et al., 2002*; *Le Marec and Lalonde, 1997*). We found that locomotor deficits in *pcd* were restricted to specific aspects of multijoint, interlimb, and whole-body coordination, while the forward trajectories of individual paws were spared (*Machado et al., 2015*). We further found that the tail movements of *pcd* mice reflected the passive consequences of limb movement (*Machado et al., 2015*). However, it remained unclear to what extent these features represented fundamental features of cerebellar ataxia, or were specific to *pcd* mice.

*Reeler* mice are a classic ataxic mutant (*Cendelin, 2014*; *Curran and D'Arcangelo, 1998*; *D'Arcangelo et al., 1999*; *D'Arcangelo et al., 1995*; *Falconer, 1951*) with an autosomal recessive mutation in the reelin gene, which is important for neural cell migration (*Beckers et al., 1994*; *Hack et al., 2002*). Its loss causes several defects, in particular aberrant localization of neurons and failure of neuronal layer formation. Several brain regions are affected, including cerebellum (*Hamburgh, 1963*; *Terashima et al., 1983*), hippocampus (*Stanfield et al., 1979*), neocortex (*Mikoshiba et al., 1980*), inferior olive (*Blatt and Eisenman, 1988*) and substantia nigra (*Kang et al., 2010*). Neuropathology in these mice is particularly striking within the cerebellum, where severe irregularities in cellular localization are also associated with corresponding aberrant synaptic connectivity between cell types, abnormal foliation, and hypoplasia. Although their locomotor kinematics and whole-body coordination have not been reported, homozygous *reeler* mutants have been described as having a severely ataxic, 'reeling' gait, with difficulties in maintaining their hindquarters upright (*Cendelin, 2014*; *Lalonde et al., 2004*; *Lalonde and Strazielle, 2019*). Like most ataxic mutants, *reelers* also exhibit poor performance in rotarod, stationary beam and water maze tests (*Lalonde et al., 2004*).

Thus, *pcd* and *reeler* mice share grossly abnormal cerebellar circuitry, but exhibit marked differences in synaptic connectivity within the cerebellum and across the brain. We wondered whether these similarities and differences on the anatomical level might be associated with similarly shared and distinct features of motor behavior. Here we analyze the locomotor behavior of *reeler* mutants and compare it quantitatively to that of the more mildly ataxic *pcd* mice (*Machado et al., 2015*). Detailed comparison of locomotor kinematics and linear discriminant analysis reveals both shared and distinct features of gait ataxia in these two mutants. This approach provides a quantitative foundation for mapping specific locomotor impairments onto distinct neuropathologies.

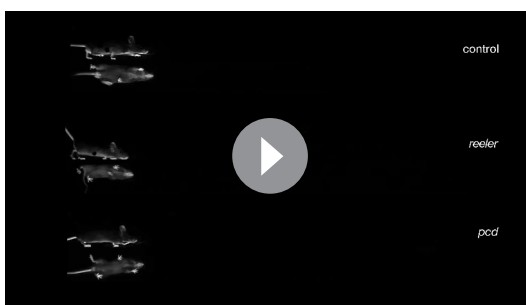

**Video 1.** Visible gait ataxia in *reeler* and *Purkinje cell degeneration* mice walking on the LocoMouse setup. Wild type (top), *reeler* (middle), and *pcd* (bottom) mice were recorded walking across the glass corridor of the LocoMouse setup. Side and bottom (via mirror reflection) views were captured by a single high-speed camera at 400fps and are shown here at 50 fps (slowed down 8x). Note the slower walking speeds of both mutants and the visible differences in their locomotor behavior.
https://elifesciences.org/articles/55356#video1

## Results

### *Reeler* mice have impaired hindlimb control and exhibit increased variability of movement

*Reeler* mice exhibited visible and severe gait ataxia when walking on the LocoMouse setup (*Video 1*). Like *pcd* mice (*Machado et al., 2015*), *reelers* were smaller and walked more slowly than control littermates (Materials and methods; *Figure 1D*). However, the locomotor phenotypes of *reeler* and *pcd* mice were clearly distinguishable by eye, with *reeler* mice appearing much more severely ataxic than the mildly ataxic *pcd* mice (*Video 1*; *Lalonde and Strazielle, 2007*; *Lalonde and Strazielle, 2019*; *Machado et al., 2015*).

We analyzed the locomotor phenotype of *reeler* mice using the quantitative framework for locomotor coordination that we established previously (*Figure 1A–C*; *Machado et al., 2015*).

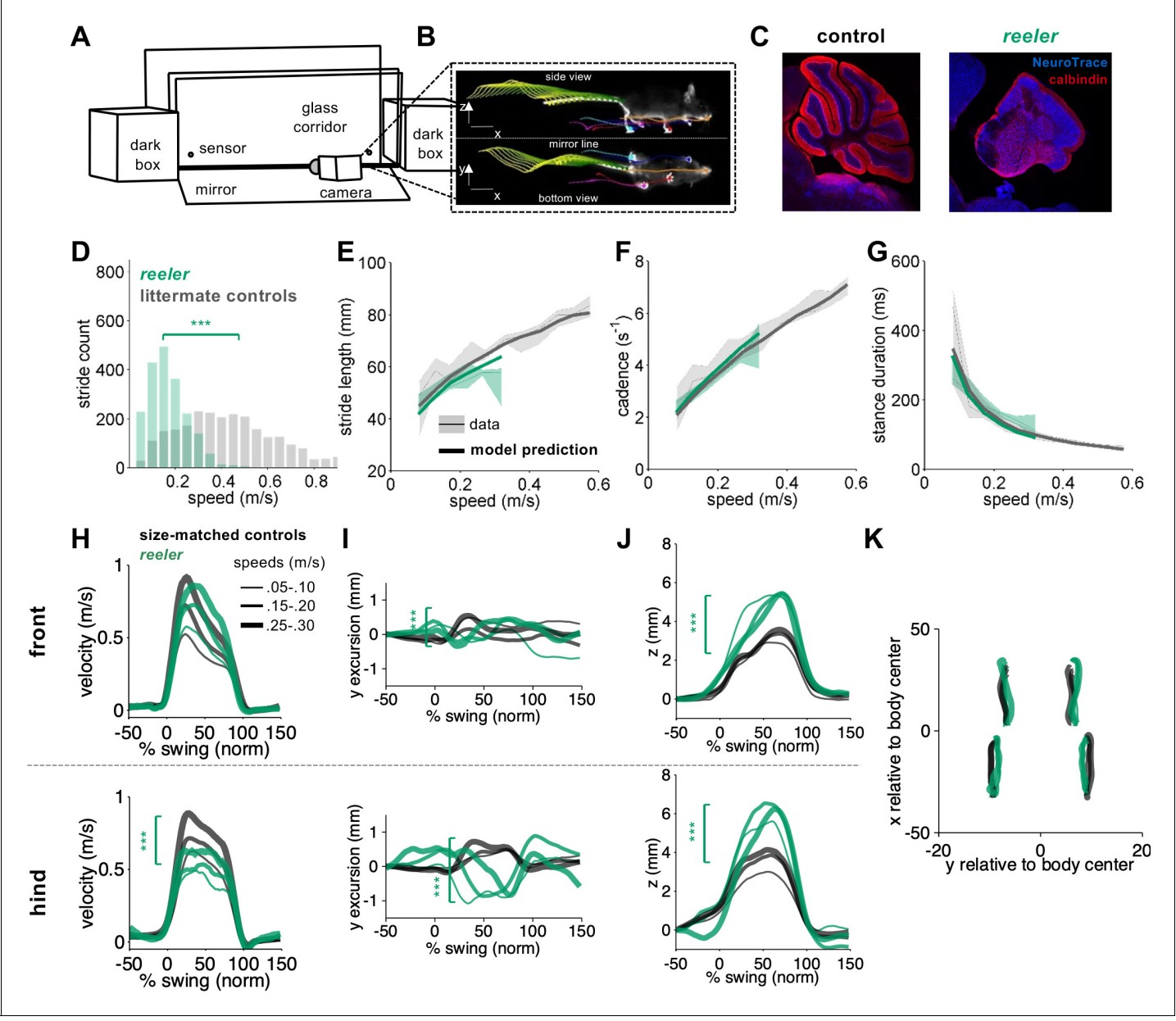

**Figure 1.** Intact forward motion of front paws, altered 3D paw trajectories, and impaired hindlimb control in *reeler*. (A) Schematic of the LocoMouse setup with two dark boxes, glass corridor, motion sensors, high speed (400fps) camera, and mirror. Mice freely cross the corridor. (B) An example of side and bottom views captured in a single via mirror reflection. Continuous tracks (in x, y, z) for nose, paws and tail segments obtained from LocoMouse tracking are plotted on top of the frame. (C) Sagittal sections of mouse cerebellum from littermate control (left) and a *reeler* mouse (right) illustrate dramatic cerebellar reorganization in *reeler*. (D) Histogram of walking speeds for *reeler* (green N = 7 mice, n = 2439) and littermate controls (grey, N = 12, n = 2515). Walking speed distributions are significantly different, *reelers* mice walk slower (ind. t-test p=<0.001***). (E–G) Stride length (E), cadence (F, 1/stride duration) and stance duration (G) of the front right (FR) paw vs walking speed for *reeler* (green) and littermates (grey). For each parameter, thin lines with shadows represent median values ± 25th, 75th percentiles. Thick lines represent the predictions calculated using the equations previously derived from the mixed-effect models described in *Machado et al., 2015*. No significant difference was observed between littermate controls and *reeler* mice (main effects: stride length: $F_{1,90}$=2.16, p=0.14; cadence: $F_{1,90}$ = 0.7, p=0.4; stance duration: $F_{1,90}$=2.97, p=0.09). (H) Average instantaneous forward (x) velocity of FR paw (top) and hind right (HR) paw (bottom), normalized to the swing phase. Line thickness represents increasing speed. *Reeler* (green), size-matched controls (black; N = 11; n = 3412). Reeler mice showed sig. higher avg. swing velocity ($F_{1,104}$ = 4.59, p=0.03), but no difference in peak inst. velocity ($F_{1,104}$ = 0.87, p=0.35). Hind paws showed lower peak velocity than size-matched controls ($F_{1,103}$ = 14.1, p=<0.0001). (I) side-to-side (y)-excursion for FR and HR paws, relative to body midline. There are changes in peak to peak trajectories for both paws (FR: $F_{1,96}$=197.4, p=<0.0001; HR: $F_{1,103}$=353.9, p=<0.0001). (J) Average vertical (z) position of FR paw (top) and HR paw (bottom) relative to ground during swing. *Reelers* mice have larger vertical movement than size-matched controls (FR: $F_{1,96}$=205.5, p=<0.0001; HR: $F_{1,103}$=11.9, p=<0.0001). (K) x-y position

*Figure 1 continued on next page*

*Figure 1 continued*

of four paws relative to the body center during swing for *reeler* and size-matched controls. There was no significant difference in width of base of support ($F_{1,101}$=2.4, p=0.12).

The online version of this article includes the following source data for figure 1:

**Source data 1.** Source data for *Figure 1*.

First, the equations we previously generated with mixed-effects linear models (*Machado et al., 2015*) to predict stride parameters based on walking speed and body size were able to accurately predict the individual stride parameters (including stride length, cadence, stance duration) of *reeler* mice (*Figure 1*. E-G), as we had previously shown for *pcd* (*Machado et al., 2015*). Moreover, the continuous forward trajectories of *reeler* front paws were similar to those of size-matched controls, across walking speeds (*Figure 1H*, top). Notably, the forward trajectories of *reeler* **hind** paws exhibited lower forward velocities compared to size and speed-matched controls (*Figure 1H*, bottom), and they had accompanying increases in swing duration (from size-matched controls = 98.78 ± 6.48 ms to *reeler* = 114.72 ± 8.21 ms). Comparison of 3D trajectories revealed clear differences between *reeler* and control mice in both the side-to-side and the vertical paw movements of both front and hind paws (*Figure 1I–K*). Finally, and perhaps surprisingly given their severe ataxia, *reeler* mice did not exhibit an increased width of base of support (*Figure 1K*).

Direct comparison of average paw kinematics in the two mutants reveals remarkable similarities (*Figure 2*). Both *reeler* and *pcd* exhibited altered 3D trajectories of all paws when compared to those of speed and size-matched controls (*Figure 2E–G*). In particular, the off-axis (side-to-side and vertical) movements of all paws showed nearly identical alterations in the two mutants (*Figure 2F, G*). In addition, forward hind paw trajectories were profoundly affected in reelers, while more subtle effects were seen on the forward movement of the front paws (*Figure 2B–E*).

In contrast to the broad similarities in *averaged* paw trajectories, there were clear differences in the *variability* of paw kinematics between *reeler* and *pcd* (*Figure 2H,I*). Despite their ataxia, neither the front nor hind limb trajectories of *pcd* mice exhibited increased variability (*Figure 2H*; *Machado et al., 2015*). Paw movement was generally more variable in *reeler*, including the forward motion of the hind paws (*Figure 2H*-bottom) and the vertical movements of both front and hind paws (*Figure 2I*-top and bottom).

## Impaired interlimb coordination and increased front paw support in *Reeler*

Mice typically walk in a symmetrical trot pattern across a wide range of walking speeds (*Figure 3A*; *Machado et al., 2015*). The normal pattern of interlimb coordination was markedly disrupted in *reeler*, due to specific and consistent changes in the phase relationship between front and hind limbs (*Figure 3B–C*). Remarkably, the alterations in front-hind limb stance phasing in *reelers* were identical on average to those of *pcd* mice; in both mutants, hind paw touch downs were delayed relative to their diagonal partners (*Figure 3B,C*; *Machado et al., 2015*). In contrast, relative left-right stance phasing of both the front and hind limbs was intact in both *reelers* and *pcd* (*Figure 3D*; *Machado et al., 2015*). Consistent with the increased variability of hind limb movements, the front-hind limb phasing was more variable on average in *reeler*, but not *pcd* (size-matched controls = 0.25 ± 0.05%, *pcd* = 0.29 ± 0.09% $F_{1,82}$ = 1.02, p=0.32, *reeler* = 0.48 ± 0.11% $F_{1,82}$ = 5.79, **p=9.5×10$^{-3}$), and left-right phasing was not more variable in either mutant (size-matched controls = 0.25 ± 0.06%, *pcd* = 0.19 ± 0.07% $F_{1,54}$ = 0.23, p=0.63, *reeler* = 0.22 ± 0.07%, $F_{1,74}$ = 0.11, p=0.75).

We also observed changes in support patterns (ie, the configuration of paws that are in stance at any given time, *Gorska et al., 1998*) in *reeler* mice. At most natural walking speeds, wildtype mice typically have a single diagonal pair of paws on the ground at any given time (*Machado et al., 2015*). *Reeler* mice exhibited an increase in 3-paw support patterns (*Figure 3E*). They also spent more time in unstable support configurations such as non-diagonal 2-paw support (*Figure 3F*) and 2-front paw supports (*Figure 3G*). This increased instability was also observed in *pcd* (*Machado et al., 2015*) and is consistent with impaired interlimb coordination rather than a simple switch to a different gait pattern (*Bellardita and Kiehn, 2015*).

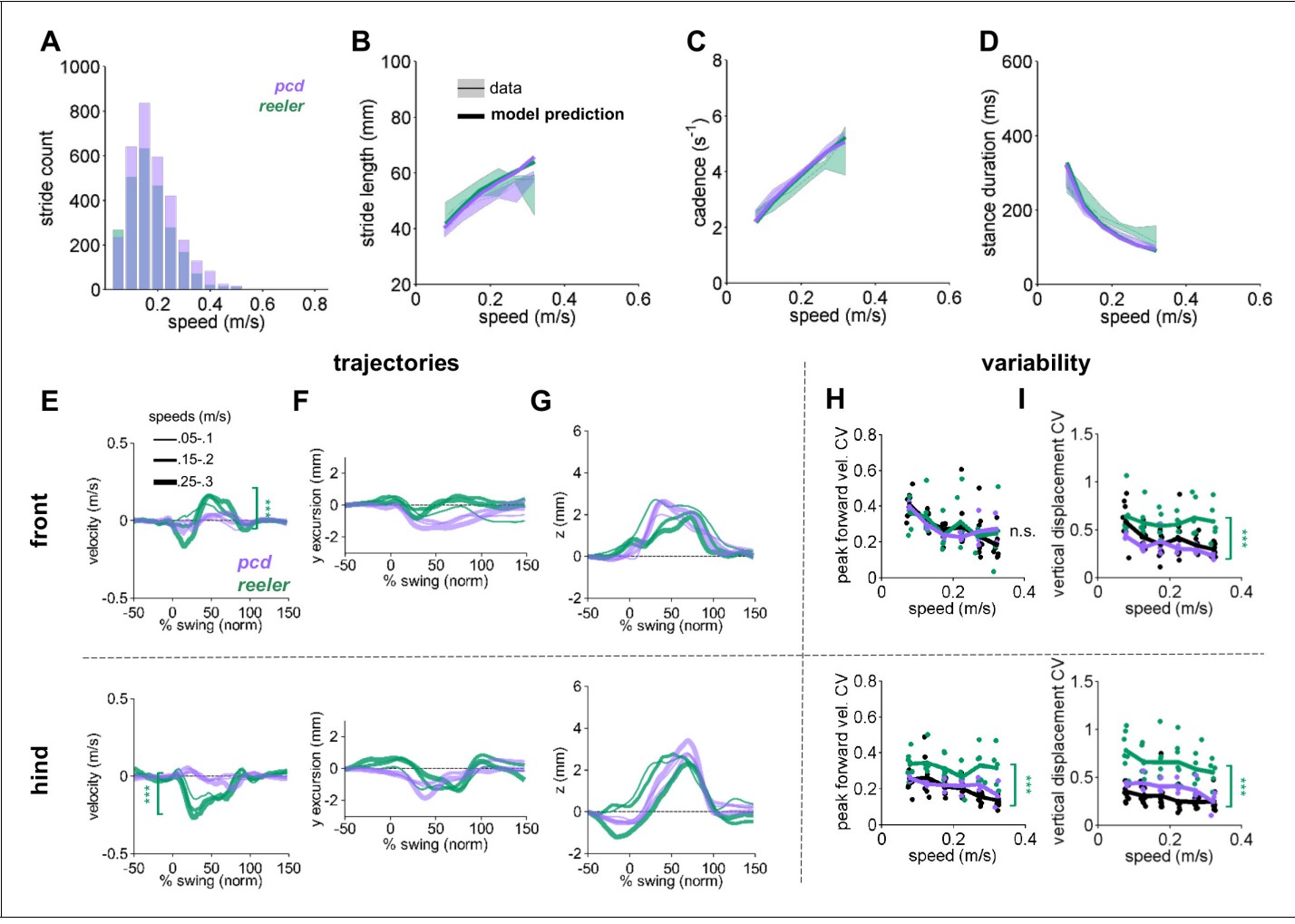

**Figure 2.** Comparison of *Reeler* and *pcd* mice reveals overall similarity of averaged paw trajectories, with additional hind limb impairments and increased variability in *reeler*. (A) Histogram of walking speeds, for *reeler* (green N = 7; n = 2439) and *pcd* (purple, N = 3; n = 3052; *Machado et al., 2015*). (B–D) Stride length (B), cadence (C), 1/stride duration) and stance duration (D) vs walking speed for *reeler* (green) and *pcd* (purple) mice. For each parameter, the thin lines with shadows represent median values ± 25th, 75th percentiles. Thick lines represent the predictions calculated based on the models derived in *Machado et al., 2015*. Reeler mice had sig. higher stride lengths ($F_{1,52}$=5.23, p=0.03). No significant differences were observed between *pcd* and *reeler* in cadence: $F_{1,55}$ = 0.01, p=0.92 or stance duration: $F_{1,52}$=0.19, p=0.66). (E–G) The differences in averaged trajectories between each mutant size and speed-matched controls are plotted for *reeler* (green) and *pcd* (purple). Line thicknesses represent increasing walking speed. (E) The peak instantaneous forward (x) velocity of FR paws (top) was sig. higher in reeler ($F_{1,8}$=50.23, p=<0.0001). Peak HR paw velocity is lower in *reeler* (bottom, $F_{1,8}$ = 6.09, p=<0.0001). (F) Differences in side-to-side (y)-excursion for FR (top) and HR (bottom) paws during swing phase, relative to body midline. No significant difference is observed in peak excursion between *pcd* and *reeler* (FR: $F_{1,8}$=4.81, p=0.06; HR: $F_{1,8}$ = 0.04, p=0.84). (G) Differences in vertical (z) trajectory of FR paw (top) and HR paw (bottom) during swing phase. No significant difference in peak z was observed between *pcd* and *reeler* mice (FR: $F_{1,8}$=0.91, p=0.37; HR: $F_{1,8}$ = 1.98, p=0.2). (H,I) Coefficient of variation (CV) for peak forward velocity (H) and vertical displacement (I) for size-matched controls, *reeler*, and *pcd*. Hind paw velocity (H, bottom; $F_{1,99}$=13, p=<0.0001) and both front (I, top; $F_{1,101}$=45.1, p=<0.0001) and hind (I, bottom; $F_{1,101}$=73.3, p=<0.0001) paw vertical movements were more variable in *reeler*.

The online version of this article includes the following source data for figure 2:

**Source data 1.** Source data for *Figure 2*.

Although the alterations in interlimb coordination were similar in *reeler* and *pcd*, there were some notable differences between the two mutants. In particular, *reeler* mice spent more time with both front paws on the ground than both controls and *pcd*, either as sole supports (*Figure 3G*), or as part of a 3 or four paw support configuration (*Figure 3H*). Interestingly, *reelers* also spent less time with both hind paws on the ground than *pcd* or control mice (*Figure 3I*). The ratio of front to hind paw double support was higher in *reeler* (front/hind = 0.43) than in *pcd* (front/hind = 0.14) and

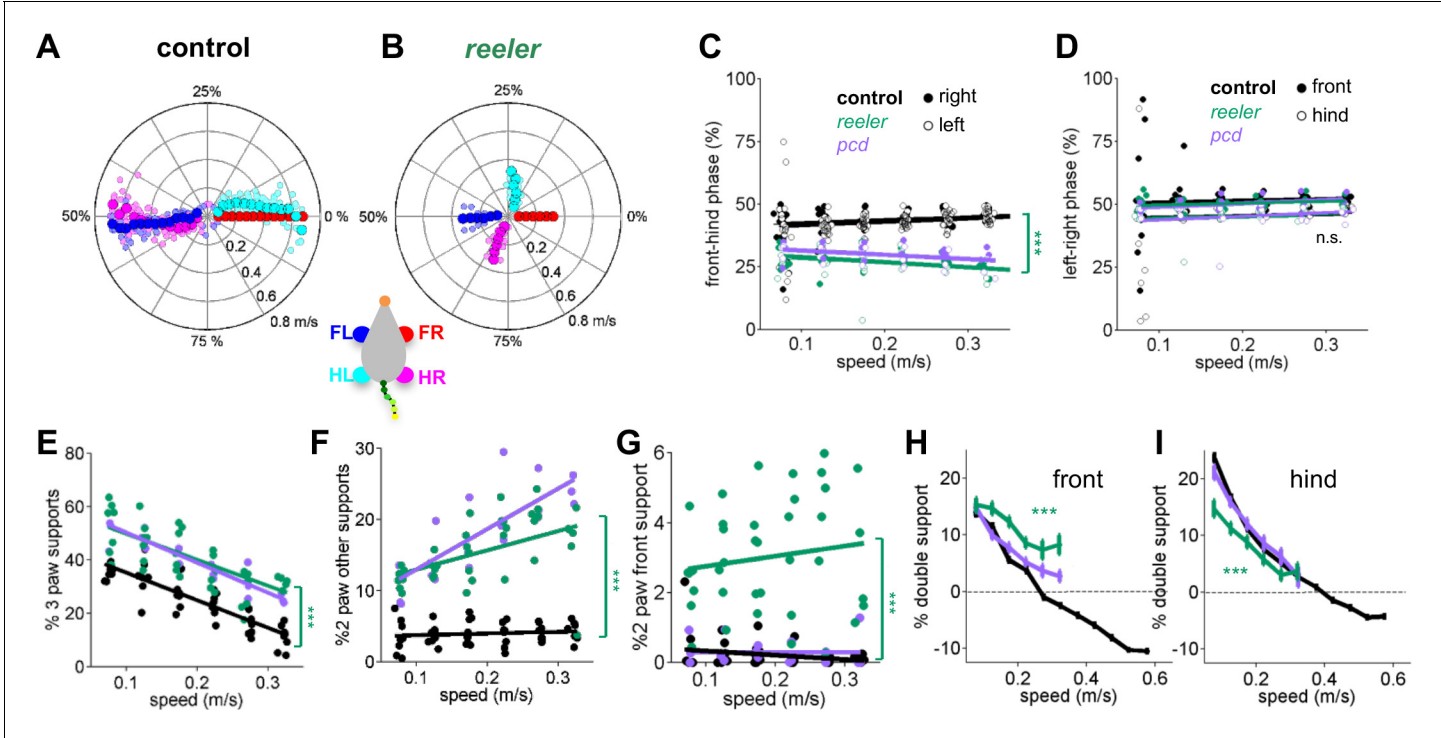

**Figure 3.** Impaired front-hind limb coordination and increased front paw support patterns in *reeler*. (A–B) Polar plots indicating the phase of the step cycle in which each limb enters stance, aligned to stance onset of FR paw (red). Radial axis represents walking speed. Limbs are color coded according to the inset; large symbols represent averages across animals and small symbols represent individual mice. (A) size-matched control mice (N = 11) and (B) *reeler* mice (N = 7). (C,D) Relative front-hind (C) and left-right (D) stance phases across walking speeds for *pcd* (purple), *reeler* (green)r hases across walking spee(black). Each circle represents one animal. Lines show fit of linear-mixed effects model for each variable. Only front-hind phase is impaired in *reeler* and *pcd* mice (front-hind phase: $F_{1,104}$ = 11.7, p=<0.0001; left-right phase: $F_{1,104}$ = 0.7, p=0.41). (E) Both *pcd* and *reeler* have a higher percentage of 3 paw supports at all speeds ($F_{1,104}$ = 115.1, p=<0.0001). (F) Non-diagonal 2-paw support configurations are increased in both *pcd* and *reeler* ($F_{1,104}$ = 28.3, p=<0.0001). (G) Only *reeler* mice show an increase in 2-front paw support configurations ($F_{1,101}$ = 207, p=<0.0001). (H,I) Average ). n increase in 2-front paw support configurations (th s one (H) and hind (I) paws of *pcd* (purple), *reelers* (green) and size-matched controls (black). *Reeler* mice have a higher % of front double support and lower % of hind double support when compared with size-matched controls (front double support: $F_{1,99}$ = 71,9, p=<0.0001; hind double support: $F_{1,103}$ = 27.2, p=<0.0001).

The online version of this article includes the following source data for figure 3:

**Source data 1.** Source data for *Figure 3*.

control mice (front/hind = 0.10). This decreased hind paw double support is likely to be a consequence of the impaired hindlimb control described in *Figure 1* and *Figure 2*.

## Impaired whole-body coordination in *reeler* reflects passive consequences of front paw motion

Like *pcd* (*Machado et al., 2015*), *reeler* mice exhibited large side-to-side oscillations of the tail and nose with respect to the stride cycle (*Figure 4A–F*, green) when compared to controls (black). Also like *pcd*, both tail (*Figure 4A,C*) and nose (*Figure 4D,F*) movements became increasingly phase-lagged relative to the locomotor cycle at faster speeds.

We previously showed that the tail and nose movements of *pcd* mice could be successfully modelled as a passive consequence of hind limb movement (*Machado et al., 2015*). A model that converted hindlimb oscillation into nose and tail trajectories using solely the geometric relationships between body parts moving with fixed time delays accurately predicted the side-to-side tail and nose movements of *pcd* mice across walking speeds.

At first glance, the similarities in averaged tail trajectories suggest that the *reeler* tail and nose movements, like *pcd*, might also reflect passive consequences of limb movement during locomotion. However, the specific phase relationships of both the tail and nose with respect to the locomotor

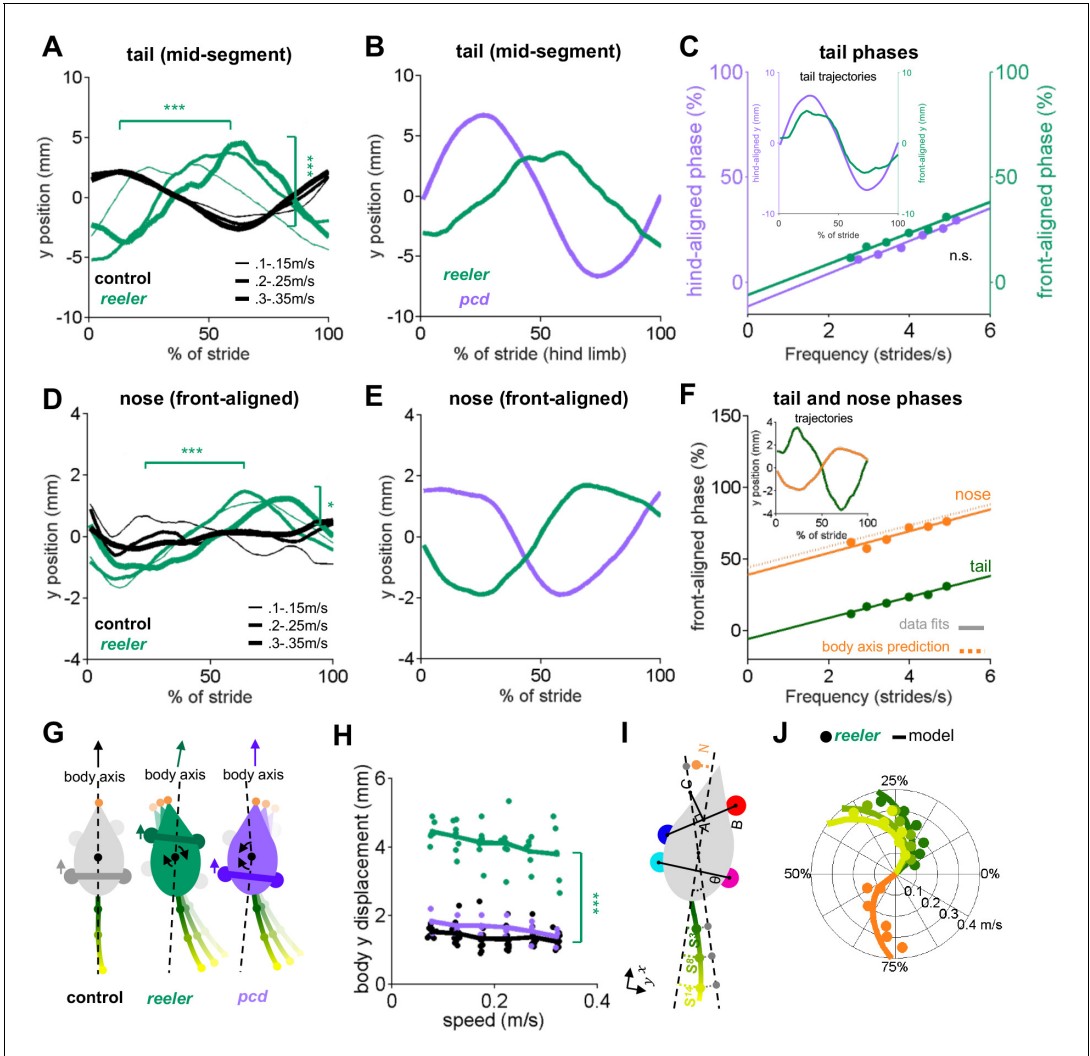

**Figure 4.** Tail and nose movements of *reeler* mice can be modeled as a passive consequence of the forward movement of the front paws. (**A**) Compared to controls (black), *reelers* (green) display larger averaged side-to-side tail oscillations, and increased phase lags with increased walking speed (from thin to thick lines) (tail amplitude: $F_{1,103}$ = 24.8, p=<0.0001; tail phase: $F_{1,104}$ = 59.2, p=<0.0001). (**B**) Different phase relationships of *reeler* (green) and *pcd* (purple) tail oscillations relative to the hind limb stride cycle (walking speed 0.25–0.3 m/s). (**C**) When aligned to the front limbs, *reeler* (green) tail phases are very close to those of *pcd* (purple) aligned to the hind limbs (lines show fits to the data; $F_{1,56}$ = 1.3, p=0.26). Inset shows the tail trajectories of *reelers* aligned to the front paws (green) and *pcd* (purple) aligned to the hind limbs (mid-tail segment for animals walking at 0.25–0.3 m/s). (**D**) *Reelers* also show larger nose oscillations and phase-lags that increase with speed increases when compared with controls (amplitude: $F_{1,104}$ = 5.1, p=0.03 ; phase:$F_{1,104}$ = 42.1, p=<0.0001). (**E**) Different phases of *reeler* nose (green) and *pcd* (purple), aligned to front paws. (**F**) The *reeler* nose (orange) is nearly perfectly out of phase with the base of the tail (green), suggesting oscillation of a single body axis (circles represent data points, solid lines show fits to the data, dashed line shows a prediction of the nose phases with respect to the same body axis as the tail. Inset shows the trajectories of the base of the tail and nose aligned to front limbs. (**G**) Interpretation of tail and nose movements observed in control (left), reeler (middle) and pcd (right) mice. (**H**) Average side-to-side (y)-excursion of the body center during strides ($F_{1,99}$=1072.5, p=<0.0001). (**I**) Geometric interpretation of the analytical model (see Materials and methods). The forward movement of front limbs (AB) is transformed into lateral oscillations of a body axis (AC). The lateral oscillations of tail and nose are then given by a time delay relative to the movement of the body axis. (**J**) Phase (relative to front limb oscillation) of nose (orange), base of the tail (dark green), mid-tail segment (intermediate green) and tip of the tail (light green), plotted as a function of walking speed. Circles represent data, lines are the predictions of the analytical model.

The online version of this article includes the following source data and figure supplement(s) for figure 4:

**Source data 1.** Source data for *Figure 4* and related supplements.
**Figure supplement 1.** Real and modeled trajectories of tail and nose in *reeler* and *pcd* mice.

cycle were dramatically different in *reeler* compared to *pcd* (**Figure 4B,E**). We wondered whether these phase differences could be accounted for by the shift in support patterns towards the front limbs in *reelers* (**Figure 3F–H**). To our surprise, simply aligning the *reeler* tail oscillations to the front limbs, rather than hind limbs (**Figure 4C**), immediately revealed that the phase relationship between the *reeler* tail and the **front** limbs was nearly identical to that of the *pcd* tail to the **hind** limbs (**Figure 4C**). In other words, the tail movements of *reeler* mice have the same quantitative relationship to front limb movement that the *pcd* tail had to the hind limbs.

Nose trajectories in *reeler* were similarly pronounced, but also phase-shifted, compared to those of *pcd* (**Figure 4D,E**). Side-to-side nose movements in *reeler* were almost perfectly out of phase with the base of the tail (**Figure 4F**), suggesting that both the tail and nose movements of *reeler* mice oscillate along a single, straight body axis with each stride (**Figure 4G**). One possible explanation for the differences in relative tail and nose phasing between *pcd* and *reeler* (**Figure 4E**) is that the shift of supports to front paws leads to the loss of a front-limb steering component in *reeler* (**Figure 4G**). Consistent with this idea, *reelers*, but not *pcd*, exhibited larger side-to-side movements of the body center while walking (**Figure 4H**, **Video 1**).

To test the idea that the nose and tail movements of *reeler* mice might reflect the passive consequence of front, rather than hind, limb movement, we built an analytical model that computed predicted lateral trajectories of the tail and nose directly from the forward-backward oscillations of the front limbs (**Figure 4I**; Materials and methods). The model and its parameters were analogous to the geometrical model described in **Machado et al., 2015**, but with a shift to the front limbs and a single body axis for *reeler* (see Materials and methods). The model output accurately reproduced the phases as well as the actual trajectories of the tail and nose of *reeler* mice, across a range of walking speeds (**Figure 4J** and **Figure 4—figure supplement 1A–C**). Notably, hindlimb-driven models, although optimal for *pcd* (**Figure 4—figure supplement 1H,I**), performed less well at predicting *reeler* tail and nose movements, even allowing for the possibility of substantially longer time delays (**Figure 4—figure supplement 1D-F**). Similarly, unlike *reeler*, a single body-axis model was not appropriate for the *pcd* data.

Thus, in *reeler*, like *pcd*, the tail and nose appear to move as a passive consequence of forward limb motion. The altered phasing in the two mutants suggests that the shift in the support patterns towards the front paws in *reeler* (**Figure 3**, and presumably resulting from impaired hindlimb control shown in **Figure 1**) causes the tail and body axis to oscillate as a passive consequence of the front, rather than hind limbs.

## Linear discriminant analysis reveals shared and specific features of gait ataxia

The results so far have described a comprehensive set of locomotor features and highlighted similarities and differences between *reeler* and *pcd* mice. Finally, we sought an unbiased way to summarize and conceptualize these findings (**Brown and de Bivort, 2018**; **Berman, 2018**; **Datta et al., 2019**). To do this we turned to linear discriminant analysis (LDA; **Fisher, 1936**; **James et al., 2013**), as a means to quantitatively distinguish the three groups of mice (wildtype, *pcd* and *reeler*; **Figure 5**).

Forty-five variables representing the various features of paw, nose, and tail movements that we measured during locomotion were extracted for this analysis (**Figure 5—figure supplement 1**). Because this relatively high-dimensional dataset contained many highly correlated variables (**Figure 5—figure supplement 2A**), we first applied principal component analysis (PCA) to account for inter-variable correlations and avoid overfitting (see Materials and methods, **Figure 5—figure supplement 2B–F**).

LDA successfully captured meaningful differences in an abstract feature subspace to separate the three genotypes based on their locomotor phenotypes across walking speeds (**Figure 5**). Unexpectedly, this analysis revealed that two distinct, orthogonal axes effectively separate controls from mutants (LD1), and the two mutants from each other (LD2) (**Figure 5**). The locomotor features contributing most strongly to these two linear discriminants thus correspond to the shared (LD1) and distinct (LD2) features of ataxia in *reeler* and *pcd* mice.

Inspection of the contributions of each gait parameter to the two LDs reveals that LD1, which separates controls from ataxic mutants, is highly influenced by variables representing 3D paw trajectories and interlimb and whole-body coordination (**Figure 5**, x-axis bar graph). The features contributing most strongly to LD2, which captured the differences in locomotor phenotype between the

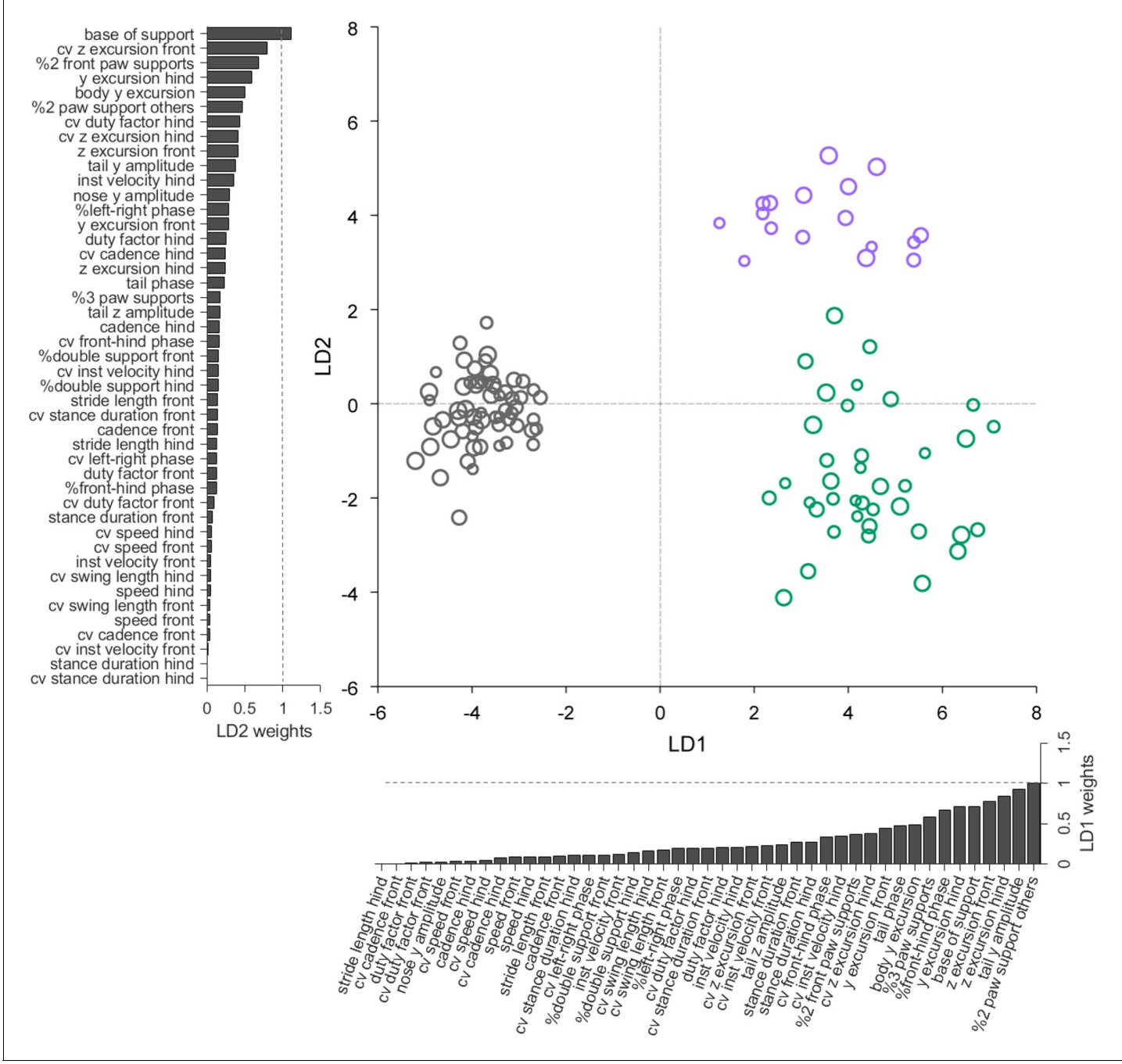

**Figure 5.** Linear discriminant analysis separates ataxic mutants and reveals shared and distinct features of gait ataxia. Linear discriminant analysis of locomotor kinematics reveals two axes, which separate ataxic mutants from controls (LD1) and from each other (LD2). Each dot represents a single animal walking at a particular speed. Faster speeds are shown with larger marker sizes. Speeds ranged from 0.05 to 0.35 m/s and were binned with a binwidth of 0.05 m/s. Size-matched controls are in grey (N = 11 for all speed bins; n = ~ 3288), reeler in green (N = 7 for 0.05–0.15 m/s; N = 6 for 0.25–0.35 m/s; n = ~ 2387), and pcd in purple (N = 3 for all speed bins except 0.25–0.30 m/s N = 2; n = ~ 3066). The bars along each axis are ranked by the contribution scores (LD coefficients) of each variable to that axis (larger bars indicate higher contributions). Features contributing strongly to LD1 (which accounts for 84% of the total between-group variance) include interlimb and whole-body coordination, as well as off-axis paw trajectories. For LD2 (which accounts for 16% of the between-group variance), they also include variability, front paw supports, and relative phasing of tail/nose movements. The online version of this article includes the following source data and figure supplement(s) for figure 5:

**Source data 1.** Source data for *Figure 5* and related supplements.
**Figure supplement 1.** Distribution of z-scored values for the gait parameters given as input to the LDA.
**Figure supplement 2.** Data correlations and variance analysis for inputs to LDA.

two mutants, included measurements of hindpaw movement (and subsequent changes in support patterns), variability, and relative phase of tail/nose movements (*Figure 5*, y-axis bar graph). Most variables relating to the averaged forward motion of individual limbs, which are largely spared in both mutants, do not strongly influence either LD.

The LDA provides a quantitative summary that captures the essential elements of the similarities and differences in locomotor behavior across groups of mice. In so doing, it shows that a high-dimensional set of gait parameters contains a subspace of mixed features in which multiple neuropathologies are represented behaviorally. The results, together with our previous description of the locomotor phenotype of *pcd* mice (*Machado et al., 2015*), reveal multijoint, interlimb, and whole-body coordination as shared, and possibly fundamental, features of locomotor ataxia.

## Discussion

When assessing motor coordination deficits in mice there is often a tradeoff between specificity and interpretability. For example, approaches such as rotarod testing provide measurable and intuitive low dimensional outputs, but lack specificity. Others, such as the CatWalk system (*Gabriel et al., 2009*) can provide many detailed measurements of locomotor behavior, but their meaning is not always readily apparent. With LocoMouse we have tried to provide both a comprehensive, quantitative description of locomotor behavior as well as a conceptual framework within which to interpret that high-dimensional data.

Here we analyzed the locomotor behavior of severely ataxic *reeler* mice and quantitatively compared it with that of *Purkinje cell degeneration* mutants (*Machado et al., 2015*). Detailed comparison of locomotor kinematics and linear discriminant analysis revealed both shared and distinct features of gait ataxia in the two mouse lines. Although the generality of the conclusions that can be reached from the analysis of two mutants is of course limited, the approach described here provides a comprehensive and quantitative way to map complex patterns of locomotor features onto partially overlapping neuropathologies. Extending it to additional models and manipulations could enable association of specific movement features with increasingly precise alterations in underlying neural circuits.

Our first finding lies in capturing specific quantitative differences in gait ataxia between *pcd* and *reeler* mice. Rather than simple differences in the level of severity in the same set of affected features, the visible gait differences in the two mutants (captured by the second linear discriminant) appear to stem specifically from an increase in movement variability and additional hindlimb involvement (*Cendelin, 2014*) in *reelers*. As a likely consequence of those deficits, analysis of support patterns and nose and tail movements suggests that *reelers*, unlike controls and *pcd*, use their front paws as their main supports while walking. This difference in support patterns fully accounts for differences in tail movements observed in the two mutants (*Figure 4*). It could also explain the inability of *reeler*s to walk in a straight line (*Video 1*, *Figure 4H*), likely because the front limbs are unable to provide both support and steering control to keep the body moving forward. Thus, while all of these features in combination contribute to the ability of human observers to visually distinguish the overall differences in walking patterns in the two mice, the quantitative LocoMouse analysis is able to distill them down to reveal fundamental underlying alterations in variability and hindlimb control.

Capturing the essential differences in locomotor control between ataxic phenotypes provides a necessary starting point for understanding the contributions of individual circuit elements to this complex, whole-body behavior. In *pcd*, the main anatomical phenotype is cerebellar, with striking degeneration of Purkinje cells that effectively removes cerebellar cortical input to the cerebellar nuclei. In contrast, *reeler* mice exhibit aberrant cell localization throughout the brain (*Terashima et al., 1983*; *Stanfield et al., 1979*; *Mikoshiba et al., 1980*; *Blatt and Eisenman, 1988*; *Kang et al., 2010*), which could also contribute to their gait abnormalities. Moreover, in *reeler*, abnormal cell migration during development results in aberrant circuit wiring that alters, rather than removes, Purkinje cell activity (*Dupont et al., 1983*; *Curran and D'Arcangelo, 1998*; *D'Arcangelo et al., 1995*; *Lalonde and Strazielle, 2019*).

Locomotor differences between the two mutants (captured by LD2 in *Figure 5*) thus likely represent a combination of specific differences in circuitry within the cerebellum itself plus extracerebellar effects. The differences in variability may be of cerebellar origin (*Walter et al., 2006*; *Medina and Lisberger, 2007*). Movement variability is often considered to be a hallmark of cerebellar ataxia, and

yet surprisingly, *pcd* mice were *less* variable than controls on several movement measures (*Machado et al., 2015*). We speculate that the opposite effects on variability in the two mutants may arise as a consequence of altered Purkinje cell output in *reeler*, compared to a lack of Purkinje cell output in *pcd* (*Medina and Lisberger, 2007*). Meanwhile, differences in hindlimb involvement, and the resulting reliance on front paw supports (*Figure 3G,H*), may be extracerebellar, perhaps resulting from cell positioning defects in *reeler* spinal cord (*Yip et al., 2000*; *Phelps et al., 2002*). Future experiments analyzing locomotor phenotypes in additional cerebellar and extracerebellar models will be crucial for establishing mechanistic links between specific features of locomotor behavior and precise alterations in underlying neural circuits.

Despite the more severe behavioral and anatomical phenotype in *reeler*, we found the overall pattern of affected locomotor features in *pcd* and *reeler* to be surprisingly similar. The first linear discriminant pulled out a set of shared impairments in multi-joint, interlimb, and whole-body coordination that may reflect core quantitative features of mouse cerebellar ataxia. We propose that these features comprise the gestalt impression of clearly cerebellar gait phenotypes that trained observers can readily identify, despite substantial variation in specific manifestations across models.

In particular, like *pcd* (*Machado et al., 2015*), the tail and nose movements of *reeler*s were also successfully modeled as passive consequences of limb movement, with a shift of supports to the front, rather than hind, paws. We had previously interpreted the pattern of coordination deficits in *pcd* mice, and particularly the passive tail oscillation, as consistent with the lack of an internal forward model that predicts and compensates for the consequences of movement (*Bastian et al., 1996*; *Ebner and Pasalar, 2008*; *Ito, 2008*; *Kennedy et al., 2014*; *Wolpert et al., 1998*). Specifically for locomotor control, a forward model can predict how the movement of one part of the body (e.g., a limb or a paw) will affect the movement of another part (e.g., another limb, or the tail), and inject a compensatory control signal to eliminate those consequences.

The idea that the cerebellum could provide a forward model for motor control is often invoked when considering the role of the cerebellum in motor learning (*Ito, 2008*; *Shadmehr and Krakauer, 2008*; *Wolpert et al., 1998*). Recently, we showed that both *pcd* and *reeler* mice were unable to learn to restore gait symmetry on a locomotor learning task that requires predictive control (*Darmohray et al., 2019*; *Morton and Bastian, 2006*; *Reisman et al., 2005*). That finding, together with the passive tail oscillations of both mutants, reinforces the idea that these animals lack the ability, provided by an intact cerebellum, to predict and compensate unintended movements during locomotion. The consistent lack of compensatory predictive mechanisms across mutants and behavioral paradigms suggests that they may represent core features of cerebellar contributions to coordinated locomotion. Extending the current findings and applying a similar approach across a broader range of circuit manipulations could become a key element in understanding how various neural components work together to control complex, whole-body behaviors.

## Materials and methods

### Animals

All procedures were reviewed and performed in accordance with the Champalimaud Centre for the Unknown Ethics Committee guidelines and approved by the Portuguese Direcção Geral de Veterinária (Ref. No. 0421/000/000/2015).

Heterozygous *reeler* (Relnrl) mice on a C57BL7/6 background were obtained from Jackson labs (#000235 B6C3Fe a/a-Relnrl/J). Data were collected from homozygous *reeler* mice (n = 9392 strides; N = 7 mice; two females; five males; 8–18 g; 35–52 days old; average weight = 13.57 ± 3.3 g) and their littermates (n = 9524 strides; N = 12 mice; eight females, four males; 12–25 g; 36–52 days old; average weight = 18.67 ± 3.5 g). Data from *Purkinje cell degeneration* mice (#0537 B6. BRAgtpbp1*pcd*/J) and additional wildtype C57BL7/6 (Jackson #000664) mice used for size-matching was previously collected and described in *Machado et al., 2015*. Mice were housed on a reversed light cycle 12 hr light/12 hr dark cycle, in standard cages with no more than five animals per cage. They had access to water and food ad libitum.

## Experimental procedures

### LocoMouse overground walking setup

The LocoMouse system for overground locomotion was composed by a glass corridor, 66.5 cm long and 4.5 cm wide with a mirror placed at 45 deg under the corridor. A single high-speed camera (AVT Bonito, 1440 × 250 pixels, 400 frames per second) recorded both bottom and side views of walking mice. Infrared sensors were used to automatically detect when mice entered and exited the corridor and trigger data collection, which was performed in LABVIEW 2012 (*Machado et al., 2015*).

### Data collection

Animals were handled and acclimated to the overground setup for several sessions before data collection. Individual trials consisted of single crossings of the corridor, in which mice walked freely between two dark boxes. No food or water restriction or reward was used. Ten to twenty-five corridor trials were collected in each session for five consecutive days. An average of n = 1342 ± 637 strides were collected per *reeler* mouse (n = 348 ± 116 strides per paw) and n = 635 ± 341 strides per littermate mouse (n = 162 ± 86 strides per animal per paw) were collected.

To track the paws, nose and tail of locomoting mice we used the previously described noninvasive, markerless LocoMouse tracking system (*Machado et al., 2015*; https://github.com/careylab/LocoMouse), with additional, subsequent updates to the tail tracking, in order to handle the more erratic tail movements of the *reeler* mice, which often left the field of view of the videos. The new tail tracking algorithm was implemented using Matlab and the Signal Processing, Image Processing and Statistics and Machine Learning toolboxes. Tail tracking started with the side view using binary thresholding followed by a skeletonization operation for finding candidate tail segments based on length and position. These points were then projected onto the bottom view starting from the distal tail segment. The bottom view image was convoluted with a hamming window with a kernel width representative of a mouse tail. Subsequent points were identified iteratively towards the proximal tail, stopping at the base of the tail. In the case of tail segmentation after skeletonization the additional step of looking for tail points towards the distal tail was taken. The tail was then divided into 15 tail points (referred to as segments) with constant Euclidian distance (in 3D) between them, similarly to *Machado et al., 2015*. Matlab code for the updated tail tracker is available at https://github.com/careylab/LocoMouse_Dev.

## Data analysis and statistics

Matlab 2012b and 2015a were used to process and analyze the data. Paw, nose and tail tracks (x,y,z) were obtained from the LocoMouse tracker (*Machado et al., 2015*). All tracks were divided in strides cycles. Stride cycle was defined as the period from stance onset to subsequent stance onset. For each stride, average walking speed was calculated. All data were sorted into speed bins (0.05 m/s bin width). Individual limb movements and interlimb coordination were calculated as follows:

### Individual limb parameters

*Walking speed*: x displacement of the body center during that stride divided by the stride duration.
Stride duration: time between two consecutive stance onsets.
*Cadence*: inverse of stride duration.
*Swing velocity*: x displacement of single limb during swing phase divided by swing duration.
*Stride length*: x displacement from touchdown to touchdown of single limb.
*Stance duration*: time in milliseconds that foot is on the ground during stride.
*Duty factor*: stance duration divided by stride duration.
*Trajectories*: (x,y,z) trajectories were aligned to swing onset and resampled to 100 equidistant points using linear interpolation. Interpolated trajectories were then binned by speed and the average trajectory was computed for each individual animal and smoothed with a Savitzky-Golay first-order filter with a 3-point window size.
*Instantaneous swing velocity*: the derivative of swing trajectory.
*Model predictions*: equations that were previously generated with mixed-effects models (*Machado et al., 2015*) to predict basic stride parameters.

## Interlimb and whole-body coordination parameters

*Base of support*: width between the two front and two hind paws during stance phase.

*Body y displacement*: y displacement of the body center during that stride.

*Stance phase*: relative timing of limb touchdowns to stride cycle of reference paw (FR). Calculated as: stance time - stance time$_{reference\ paw}$/stride duration.

*Supports*: Support types were categorized by how many and which paws were on the ground, expressed as a percentage of the total stride duration for each stride. Paw support categories include 3-paw, 2-paw diagonal, 2-paw other/non diagonal (homolateral and homologous), and 2-paw front (only)supports.

*Double support for each limb* is defined as the percentage of the stride cycle between the touch down of a reference paw to lift-off of the contralateral paw. Because at higher speeds (running), the opposing limb lifts off before the reference paw touches down, we included negative double support by looking backwards in time, up to 25% of the stride cycle duration. Positive values of double support indicate that contralateral lift-off occurred after reference paw touch down, and negative values indicate that contralateral lift-off occurred before reference paw touch down. Note that front paw double support percentages include 2-paw front (only) support patterns as well as 3 and 4-paw support patterns in which both front paws were on the ground.

*Tail and nose phases*: For each speed bin we correlate the stridewise tail and nose trajectories with the trajectory given by the difference between the forward position of the right paw and the forward position of the left paw (also normalized to the stride). We do this both for front limbs (for the analysis of *reeler* mice) and hind limbs (for the analysis of previous *pcd* data). The phase is then calculated by the delay in which this correlation is maximized.

*Tail and nose peak-to-peak amplitude*: the change between peak (highest amplitude value) and trough (lowest amplitude value) in y or z during a stride duration.

*Variability*: All variability analyses were based on coefficients of variation (CV).

## Geometric model of the tail and nose

The analytical model of the nose and tail was a simpler version of our previously described geometric model (*Machado et al., 2015*). The current model transforms the forward movements of the front limbs into predicted lateral oscillations of tail and nose. The model is described by the equation:

$$y_t^s = G^s(X_{t-Ds}^r - X_{t-Ds}^l) \text{ and } y_t^n = G^n(X_{t-Dn}^l - X_{t-Dn}^r)$$

where $y_t^s$ and $y_t^n$ are the lateral positions of tail segment $s$ and nose, respectively; $G^s$ and $G^n$ are gains, obtained from fitting the data, that transform the limb oscillation amplitude to the amplitude of tail segment $s$ and nose movements, respectively; $x_t^r$ and $x_t^l$ are the positions of front right and front left limbs at time t obtained from average trajectories of limb movements during strides at different speeds. $Ds$ and $Dn$ are the delays of tail segment $s$ and nose, where the delay $D1$ is the delay of the base of the tail obtained by fitting the data.

As in *Machado et al., 2015*, delays between subsequent tail segments decreased according to the equation:

$$Ds = -0.23^*s + 3.97$$

where $s$ is the segment number, starting at the base of the tail. The delay of the nose was the same as the delay of the base of the tail (e.g. $Dn = D1$).

## Principal component and linear discriminant analyses

The dataset consisted of a matrix of 109 observations of 45 features. Each observation was data from one mouse locomoting at a certain speed (binned) and features are z-scored gait parameters. LDA assumes independence within the feature space, which we knew to be violated due to the high speed-dependence of many gait features (*Machado et al., 2015*). Therefore, PCA was applied to address inter-variable correlation and avoid overfitting in LDA. PCA was performed by eigenvalue decomposition of the data covariance matrix. The first 10 PCs explained 88% of the variance and the data projected onto these 10 PCs were used as input to the LDA. The end contributions of the initial gait parameters to the two LD axes were obtained by multiplying the PCA mapping by the LDA

mapping. LDA output is presented for each speed bin to verify that the pattern of differences across groups was captured, across all walking speeds.

## Statistical analyses

All statistics can be found in Table S1. Statistical analyses were done in Matlab with the Statistics toolbox. An independent samples t-test was used to test for differences in walking speed distributions (*Figure 1D* and *Figure 2A*). For all other gait parameters, analysis was performed on animal averages binned by speed using mixed-effects models (*Bates et al., 2013*). Fixed-effects terms included speed and genotype; animals were included as random terms. We report F statistics from mixed ANOVAs with Satterthwaite degrees of freedom correction. Differences were considered significant at $*p < 0.05$, $**p < 0.01$, and $***p < 0.001$; asterisks report main effects of genotype.

# Acknowledgements

We thank Tracy Pritchett for maintenance of mouse lines and help with histology and imaging and João Fayad for optimization and maintenance of the LocoMouse tracker. Olivia Carmo assisted with data acquisition and Alex Azinheira made the video. We are grateful to the members of Carey Lab and the Champalimaud Neuroscience Program for helpful discussion.

This work was supported by a Howard Hughes Medical Institute International Early Career Scientist Grant #55007413 (to MRC), European Research Council Starting Grant #640093 (to MRC), FCT grant FCT-PTDC/MED-NEU/30890/2017 (to MRC), Congento LISBOA-01–0145-FEDER-022170, and fellowships from the Portuguese Fundação para a Ciência e a Tecnologia: SFRH/BD/51210/2010 (to ASM), SFRH/BD/86265/2012 (to DMD), PD/BD/141643/2018 (to DFD), and SFRH/BPD/119404/2016 (to HGM).

# Additional information

## Competing interests

Megan R Carey: Reviewing Editor, eLife. The other authors declare that no competing interests exist.

## Funding

| Funder | Grant reference number | Author |
| --- | --- | --- |
| H2020 European Research Council | Starting Grant 640093 | Megan R Carey |
| Howard Hughes Medical Institute | International Early Career Scientist 55007413 | Megan R Carey |
| Fundação para a Ciência e a Tecnologia | FCT-PTDC/MED-NEU/30890/2017 | Megan R Carey |
| Fundação para a Ciência e a Tecnologia | SFRH/BD/51210/2010 | Ana S Machado |
| Fundação para a Ciência e a Tecnologia | SFRH/BPD/119404/2016 | Hugo G Marques |
| Fundação para a Ciência e a Tecnologia | PD/BD/141643/2018 | Diogo F Duarte |
| Fundação para a Ciência e a Tecnologia | SFRH/BD/86265/2012 | Dana M Darmohray |

The funders had no role in study design, data collection and interpretation, or the decision to submit the work for publication.

## Author contributions

Ana S Machado, Conceptualization, Data curation, Software, Formal analysis, Investigation, Visualization, Methodology, Writing - original draft, Writing - review and editing; Hugo G Marques,

Conceptualization, Formal analysis, Investigation, Visualization, Methodology, Writing - original draft, Writing - review and editing; Diogo F Duarte, Conceptualization, Software, Formal analysis, Visualization, Writing - original draft, Writing - review and editing; Dana M Darmohray, Conceptualization, Software, Formal analysis, Validation, Investigation, Visualization, Methodology, Writing - original draft, Writing - review and editing; Megan R Carey, Conceptualization, Resources, Supervision, Funding acquisition, Validation, Visualization, Methodology, Writing - original draft, Project administration, Writing - review and editing

### Author ORCIDs
Hugo G Marques https://orcid.org/0000-0002-8709-4841
Megan R Carey https://orcid.org/0000-0002-4499-1657

### Ethics
Animal experimentation: All procedures were reviewed and performed in accordance with the Champalimaud Centre for the Unknown Ethics Committee guidelines and approved by the Portuguese Direcção Geral de Veterinária (Ref. No. 0421/000/000/2015).

### Decision letter and Author response
Decision letter https://doi.org/10.7554/eLife.55356.sa1
Author response https://doi.org/10.7554/eLife.55356.sa2

## Additional files

### Supplementary files
• Supplementary file 1. Statistics for *Figures 1–4*. To test the differences, or not, between genotypes (reelers and control) across speeds we used an independent samples t-test and mixed-effects models (*Bates et al., 2013*). In this table, we report F statistics from mixed ANOVAs with Satterthwaite degrees of freedom correction. Differences were considered significant at *$p<0.05$, **$p<0.01$, and ***$p<0.001$.

• Transparent reporting form

### Data availability
Source data have been provided for all Figures.

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
