## [Decision Letter]

[Editors’ note: the authors submitted for reconsideration following the decision after peer review. What follows is the decision letter after the first round of review.]

Thank you for submitting your work entitled "Shared and specific signatures of locomotor ataxia in mutant mice" for consideration by *eLife*. Your article has been reviewed by three peer reviewers, and the evaluation has been overseen by a Reviewing Editor and a Senior Editor. The reviewers have opted to remain anonymous.

Our decision has been reached after consultation between the reviewers. Based on these discussions and the individual reviews below, we regret to inform you that your work will not be considered further for publication in *eLife*.

The work seems like a useful extension of Machado et al., 2015, from a benchmarking perspective – this paper helps to show that LocoMouse "works" at least in the sense of being able to identify commonalities and differences between mutants and to make semantic sense of those similarities and differences. However, after extensive consultation the consensus was reached that the paper does not contain enough new and impactful science for an advance on Machado et al., 2015, especially given reeler data in Darmohray et al., 2019, and the methodology used is that previously used by the authors, albeit in a different context and with a different mutant. The attempt to tie the behavioral defects quantified to circuit level defects were not convincing.

Reviewer #1:

This manuscript is a very well written manuscript describing the differences and similarities in the ataxic leg movements during walking in two mutant mouse models that have cerebellar neuropathy. Namely these mice are the Purkinje Cell Generation mice and the *Reeler* mice. They did a nice set of recordings and data analysis. However, this manuscript is purely descriptive, without any conceptual conclusion. I don't think this work, although well done, is not appropriate for *eLife*.

Reviewer #2:

We are still at the proof-of-concept phase for computational neuroethology. One goal of these methods is to identify relationships between complex patterns of behavior by – in a task or context specific manner – balancing (as the discussion says) specificity (e.g., high dimensional measures) with interpretability (e.g., a conceptual framework). This paper puts this hypothesis to the test using the LocoMouse framework developed by this lab to compare ataxic phenotypes among *pcd* and *reeler* mice. Here the authors do a very nice job of demonstrating that – like the *pcd* mice – many of the apparent phenotypes in the *reeler* mouse are a simple consequence of the slowed speed and smaller size of the mutants. In addition, the observations that phenotypes are now largely found in the front paws – and that tail and nose dynamics can be predicted from this effect – effectively distinguishes the two mutants. The data are clear, and the findings of significant import and utility to those interested in ataxias and in motor control.

Given that the technology and analytics associated with LocoMouse are at this point well vetted, I have no specific comments or concerns about the data in the paper. I would like only a few small adjustments. It would be nice to move some of the data from Figure 1—figure supplement 1 into main Figure 1 (or maybe to make Figure 1—figure supplement 1 a new Figure 2) so that the baseline comparisons between the mutants is more front stage in the figures. In addition, it would be clarifying to generate some simplified graphs (maybe focused on specific speeds) that, in at least some cases, enable comparisons of *reeler*, *pcd* and controls all at once. The compromises used in Figure 3 could work here (where you have adjacent panels with two of the three comparisons), but on the whole it would be generally nice to see more direct comparisons across the data in the main figures. Finally, I am left with a sense of the power of the approach to both relate and distinguish, but the one thing that isn't made completely explicit is why *reeler* mice are so much more apparently ataxic when you just look at them. Is this largely the collective consequence of the "high-stepping" phenotype, the increased variability and the steering phenotype? There is obviously not a simple answer to this, but I would appreciate some discussion of what might account for the fact that the *reeler* mice look worse by eye, and how quantitative measures might highlight or obscure the gestalt we have as researchers inspecting our mice.

Reviewer #3:

The authors use established methods from their lab to compare locomotor kinematics in wildtype mice and two mutant mouse lines: *pcd* and *reeler*. This work builds on some of their impressive previous work where they already showed the results for the *pcd* mice (Machado et al., 2015), as well as evaluations of both *pcd* and *reeler* mice in a different paradigm testing locomotor kinematics, the split-belt treadmill (Darmohray et al., 2019). With the current manuscript, the authors aim to provide a quantitative foundation for mapping specific locomotor impairments onto distinct neuropathologies. The experiments seem to be competently performed, and the manuscript is well written. However, the limitation of the current manuscript is that it does not provide much new insight to the analysis of body part kinematics which hasn't already largely been covered in their previous work.

The main concerns are:

1) Lack of novelty compared to their previous papers (Machado et al., 2015; Darmohray et al., 2019). Basic locomotor kinematics of both *pcd* and *reeler* mice (as shown here in Figures 1-3) are already described in these papers. Figure 4 does contain new information, although it largely depends on the results from Figure 2. Similar observations to those in Figure 2 of the current manuscript, e.g. different paw supports for the two mutant lines compared to controls, have already been shown in Figure 3D and H of Darmohray et al., 2019, albeit in a different behavioral paradigm.

2) The authors claim to provide a quantitative foundation for mapping specific locomotor impairments onto distinct neuropathologies. However, there is not enough biological validation for the claim that the data presented in the current manuscript can link behavioral phenotypes with underlying neural circuits, as there is no detailed description of the anatomical differences between the two mutant lines within the cerebellum. Certainly the found behavioral differences between the two are not linked to their neural circuits. The authors would need to revise their conclusions based on the data they present here.

3) An attempt to possibly rescue the phenotype would help to improve the biological understanding of which behavioral phenotypes are linked to specifically which anatomical underlying structures. I.e., is it possible to intervene in one of the animal models, so that their phenotype switches on the LD1 axis presented in Figure 4 to the side of the control mice? Such conclusions would then truly link behavioral phenotypes to neural circuits, and would help to significantly improve the biological understanding of mouse locomotor kinematics.

[Editors’ note: further revisions were suggested prior to acceptance, as described below.]

Thank you for submitting your article "Shared and specific signatures of locomotor ataxia in mutant mice" for consideration by *eLife*. Your article has been reviewed by three peer reviewers, and the evaluation has been overseen by Ronald Calabrese as the Senior Editor and Reviewing Editor. The reviewers have opted to remain anonymous.

The reviewers have discussed the reviews with one another and the Reviewing Editor has drafted this decision to help you prepare a revised submission.

Summary:

In this manuscript, Machado and colleagues use LocoMouse, their behavioural tracking software package published previously as part of a study in *eLife* (Machado et al., 2015), to reveal similarities and differences in the locomotor kinematic deficits of two mouse models of ataxia, specifically *pcd* and *reeler* mice. A behavioural description of locomotion in *pcd* mice was already included in Machado et al., 2015, so the additional data provided here are for *reeler* mice. The analyses provided reveal that some locomotor parameters are similarly impaired in these two mutant models while other parameters are unique to each mutant.

The reviewers found considerable merit in this advance but have further revisions they recommend that can be accomplished quickly and evaluated by the RE. Two of the reviewers thought that further experiments were needed to sustain the authors claims but realizing that this advance was considerable recommend revision in consultation.

Reviewer 3:

"The analyses provided reveal that some locomotor parameters are similarly impaired in these two mutant models while other parameters are unique to each mutant. While the additional findings presented in this study are sound, it is unclear how much we can generalize from analysis of one additional ataxia model, particularly because the neuropathology in these two models is quite different. Without a deeper evaluation of the generality and specificity of locomotor deficits in mouse models of ataxia, this study only provides a relatively slim additional advance over their previous study, and the more general conclusions they draw are not sufficiently justified. I have the following suggestions for strengthening the present manuscript. The authors show nicely in Figure 5 that combinations of behavioral features can be used to distinguish not only wild type from mutant mice but also *pcd* and *reeler* mice from each other. The authors' interpretation of these results is that the locomotor features that commonly distinguish wild type animals from the two mutants have a common underlying neuropathological origin (i.e. a cerebellar deficit) while the features that distinguish *pcd* mice from *reeler* mice reflect differences between these two mutants (i.e. absence of Purkinje cell output vs. miswiring in the cerebellum plus the additional extracerebellar defects in *reeler* mice). This is an important idea that could be bolstered with some additional data/analysis: 1. To show that aberrant cerebellar function is the root cause of the variability along LD1, the authors could use perturbation experiments to specifically affect locomotion by manipulating activity in interpositus-projecting Purkinje cells. This could be done acutely (as in Sarnaik and Raman, 2018) and/or chronically (as in Darmohray et al., 2018). 2. To test if LD1 does indeed distinguish wild type from ataxic mice and LD2 (or higher dimensions) distinguish different types of ataxic mice from each other, the authors could analyze locomotion in other models of ataxia that have different neuropathologies."

Reviewer 2:

"On a related note, a major claim is that the LD analysis defines a core set of behavioral features associated with cerebellar dysfunction. These are: impairments in multijoint, interlimb, and whole-body coordination, as well as in predictive mechanisms (i.e. forward models). It is true that these behavioral impairments are observed in both *pcd* and *reeler* mice, which share cerebellar neuropathology. However, it is possible that pathology in other brain circuits (basal ganglia, for example) may also be associated with some of the same behavioral impairments observed in the *pcd* and *reeler* mice, which would indicate that these behavioral impairments are not unique to the cerebellum after all. The impact of the paper, and the degree to which it is possible to claim that particular abnormalities are hallmarks of cerebellar dysfunction, would be much higher if the LD analysis could be repeated in mice with gait abnormalities caused by non-cerebellar dysfunction: Amende et al., 2005. Gait dynamics in mouse models of Parkinson's disease and Huntington's disease. If it is not feasible at this time to use the Locomouse system to analyze gait abnormalities in the mouse models in the Amende paper, the paper should at least discuss some of the limitations in terms of interpreting results as capturing behavioral hallmarks of cerebellar dysfunction. The distinction is between the impairments being "specific" markers of cerebellar dysfunction (which is supported by the data) vs "unique" markers of cerebellar dysfunction (which is not supported by the data). In other words, if a mouse has cerebellar dysfunction then we would be able to predict, based on the current paper, some of the abnormalities we may expect. On the other hand, if a mouse displays some of the abnormalities, it would not be possible to assume, based on the current results, that the underlying neuropathology is cerebellar."

In light of these reservations, the authors should scale back some of their claims and discuss the limitations of the analysis more.

---

## [Author Response]

[Editors’ note: The authors appealed the original decision. What follows is the authors’ response to the first round of review.]

Reviewer #1:This manuscript is a very well written manuscript describing the differences and similarities in the ataxic leg movements during walking in two mutant mouse models that have cerebellar neuropathy. Namely these mice are the Purkinje Cell Generation mice and the Reeler mice. They did a nice set of recordings and data analysis. However, this manuscript is purely descriptive, without any conceptual conclusion. I don't think this work, although well done, is not appropriate for eLife.

The current manuscript was submitted to *eLife* as a Research Advance – a format unique to *eLife* in which a paper is closely associated with a previous *eLife* paper. In this case, the current manuscript describing *reeler* mice is a follow up of our 2015 paper that developed the LocoMouse analysis framework to quantify the gait impairments of *Purkinje cell degeneration* mice (Machado et al., 2015). Yes, the data in both papers are descriptive. But this does not mean that they have no conceptual conclusion. Many researchers have told us that our 2015 paper fundamentally transformed the way they think about mouse locomotion and the role of the cerebellum. The current manuscript provides solid evidence for some of the ideas we could only speculate about then, as well as capturing shared and specific features of two forms of gait ataxia in a broadly applicable way, for the first time. We have revised the paper, with substantive changes to the Abstract, Introduction, and Discussion, to make our motivation and conceptual points more clear. In particular, the Discussion now highlights conceptual conclusions that are only made possible by taking both of Machado et al., 2015 and the current study into account.

Reviewer #2:We are still at the proof-of-concept phase for computational neuroethology. One goal of these methods is to identify relationships between complex patterns of behavior by – in a task or context specific manner – balancing (as the discussion says) specificity (e.g., high dimensional measures) with interpretability (e.g., a conceptual framework). This paper puts this hypothesis to the test using the LocoMouse framework developed by this lab to compare ataxic phenotypes among pcd and reeler mice. Here the authors do a very nice job of demonstrating that – like the pcd mice – many of the apparent phenotypes in the reeler mouse are a simple consequence of the slowed speed and smaller size of the mutants. In addition, the observations that phenotypes are now largely found in the front paws – and that tail and nose dynamics can be predicted from this effect – effectively distinguishes the two mutants. The data are clear, and the findings of significant import and utility to those interested in ataxias and in motor control.Given that the technology and analytics associated with LocoMouse are at this point well vetted, I have no specific comments or concerns about the data in the paper. I would like only a few small adjustments. It would be nice to move some of the data from Figure 1—figure supplement 1 into main Figure 1 (or maybe to make Figure 1—figure supplement 1 a new Figure 2) so that the baseline comparisons between the mutants is more front stage in the figures.

We thank the reviewer for this suggestion. We have converted Figure 1—figure supplement 1 into a main figure (now Figure 2).

In addition, it would be clarifying to generate some simplified graphs (maybe focused on specific speeds) that, in at least some cases, enable comparisons of reeler, pcd and controls all at once. The compromises used in Figure 3 could work here (where you have adjacent panels with two of the three comparisons), but on the whole it would be generally nice to see more direct comparisons across the data in the main figures.

We agree. What makes this tricky is the need to both speed and size-match the data from the two mutants with the controls. In Figure 1—figure supplement 1 (now Figure 2), we had accomplished that by plotting the differences between each mutant and their respective size-matched controls, but this does not work for every feature. To do our best to satisfy this reviewer’s request, which is similar to some of the points raised by reviewer 3, we have a) included Figure 1—figure supplement 1 as a main figure (now Figure 2), and b) included Summary plots like those from Figure 2 for the 43 features that provided inputs to the LDA, in the new Figure 5—figure supplement 1.

Finally, I am left with a sense of the power of the approach to both relate and distinguish, but the one thing that isn't made completely explicit is why reeler mice are so much more apparently ataxic when you just look at them. Is this largely the collective consequence of the "high-stepping" phenotype, the increased variability and the steering phenotype? There is obviously not a simple answer to this, but I would appreciate some discussion of what might account for the fact that the reeler mice look worse by eye, and how quantitative measures might highlight or obscure the gestalt we have as researchers inspecting our mice.

We thank the reviewer for this suggestion and have expanded our discussion of this issue, and of the relationship between hindlimb involvement, variability, and front paw supports, along with their possible neural basis, in the Discussion.

Reviewer #3:The authors use established methods from their lab to compare locomotor kinematics in wildtype mice and two mutant mouse lines: pcd and reeler. This work builds on some of their impressive previous work where they already showed the results for the pcd mice (Machado et al., 2015), as well as evaluations of both pcd and reeler mice in a different paradigm testing locomotor kinematics, the split-belt treadmill (Darmohray et al., 2019). With the current manuscript, the authors aim to provide a quantitative foundation for mapping specific locomotor impairments onto distinct neuropathologies. The experiments seem to be competently performed, and the manuscript is well written. However, the limitation of the current manuscript is that it does not provide much new insight to the analysis of body part kinematics which hasn't already largely been covered in their previous work.

We thank the reviewer for the nice comments about our previous work. Below, we clarify some misunderstandings about what we have and have not shown before. We also note that this paper’s tight relationship to Machado et al., 2015 is precisely why we submitted this paper as a Research Advance – a format, unique to *eLife*, that allows a manuscript to be formally associated with a previous *eLife* paper.

The main concerns are:1) Lack of novelty compared to their previous papers (Machado et al., 2015; Darmohray et al., 2019). Basic locomotor kinematics of both pcd and reeler mice (as shown here in Figures 1-3) are already described in these papers. Figure 4 does contain new information, although it largely depends on the results from Figure 2. Similar observations to those in Figure 2 of the current manuscript, e.g. different paw supports for the two mutant lines compared to controls, have already been shown in Figure 3D and H of Darmohray et al., 2019, albeit in a different behavioral paradigm.

First, yes, the *pcd* data were published in Machado et al., 2015 (which is part of why we are submitting this as a Research Advance). But all of the *reeler* data in this paper is entirely new. The statement “Basic locomotor kinematics of both *pcd* and *reeler* mice (as shown here in Figures 1-3) are already described in these papers” is just not true.

That claim appears to be supported by the statement “Similar observations to those in Figure 2 of the current manuscript, e.g. different paw supports for the two mutant lines compared to controls, have already been shown in Figure 3D and H of Darmohray et al., 2019, albeit in a different behavioral paradigm.” We know that the locomotor terminology can be confusing, and so here we aim to clarify an unfortunate misunderstanding. Darmohray et al., 2019 dealt exclusively with locomotor learning. Figure 3D and H of that paper quantified the extent of spatial vs. temporal components of learning in those mutants. Changes in double support symmetry after learning were one way that we assessed the temporal component of learning, but that measure has no correspondence to any parameter reported in the current manuscript. We invite reviewers to consider of Figure 3D and H of Darmohray et al. alongside Figure 2H from our current manuscript. Immediate inspection of these two plots reveals that while they both contain the words “% double support”, they contain entirely different data. While the *reeler* datapoints hover around 0 in the learning experiments (because there was no temporal learning in these mice), the opposite is true in the current manuscript – the *reelers* never approach 0 double support.

The reason for this discrepancy is that double support aftereffects, which measure learning, are a completely different measure from double support itself. Given that *reeler* mice were not included at all in Machado et al., 2015, and that the only *reeler* data in Darmohray et al., 2019 were a few datapoints concerned only with learning, there is simply no basis for the claim that we have previously reported the locomotor kinematics of *reeler* mice, or that the *reeler* data in the first 3 figures of this paper are not new.

To avoid additional confusion, we have added a statement in the Introduction to clarify that “the locomotor kinematics and whole-body coordination of *reeler* mice have not been reported”.

Finally, the reviewer acknowledges that “Figure 4 does contain new information, although it largely depends on the results from Figure 2.” In fact, Figure 4 incorporates results from all figures of this paper and those of Machado et al., 2015, in order to synthesize this vast amount of locomotor analysis into a digestible format. This approach is entirely new and, as we discuss in the Discussion, not only does it allow us to capture shared and specific features of gait ataxia here, it will form an essential component of our ability to extract meaning from the kinds of behavioral quantification that we first described in Machado et al., 2015 and that have become much more common since.

2) The authors claim to provide a quantitative foundation for mapping specific locomotor impairments onto distinct neuropathologies. However, there is not enough biological validation for the claim that the data presented in the current manuscript can link behavioral phenotypes with underlying neural circuits, as there is no detailed description of the anatomical differences between the two mutant lines within the cerebellum. Certainly the found behavioral differences between the two are not linked to their neural circuits. The authors would need to revise their conclusions based on the data they present here.

We sincerely apologize if we mistakenly gave the impression that we were trying to link specific aspects of the altered cerebellar circuitry to specific aspects of behavior. This was completely unintentional! We have made a number of revisions to the text to more clearly convey our actual meaning, including:

– Major changes to Abstract and Introduction to provide additional context and motivation for the current study .

– Changes in language throughout the paper to favor use of “structures”,

“neuropathologies”, “anatomical features”, and similar in place of “circuitry”, “circuits”, etc.

– A clarified and extended discussion of how we think the gait alterations we observe may related to gross pathology (related to the final comment of reviewer 2).

We hope that these changes have helped clarify that rather than claiming to have mapped specific locomotor features onto specific circuit abnormalities within the cerebellum, we are trying to demonstrate the ability to map shared and specific gait abnormalities onto shared and specific anatomical features. The shared locomotor phenotypes between the two mutants are almost by definition cerebellar, for example, as the neuropathology in *pcd* is largely specific to the cerebellum (particularly at the youngest ages we tested in Machado et al., 2015) and that is the only overlapping anatomical feature between the two lines. We speculate in the Discussion (and have now expanded this section) about possible anatomical foundations for the differences between the two mutants.

Finally, although we hope it is now clear that we are not attempting to make specific claims about their altered cerebellar circuitry giving rise to specific behavioral differences (other than speculating in the Discussion about the difference between a lack of Purkinje cell output (*pcd*) vs. aberrant Purkinje cell output (*reeler*)), we note that *reeler* and *pcd* are two of the bestdescribed mutants in the literature – their neuropathologies within the cerebellum and beyond have been described and replicated in countless papers, many of which we cite and describe in some detail here.

3) An attempt to possibly rescue the phenotype would help to improve the biological understanding of which behavioral phenotypes are linked to specifically which anatomical underlying structures. I.e., is it possible to intervene in one of the animal models, so that their phenotype switches on the LD1 axis presented in Figure 4 to the side of the control mice? Such conclusions would then truly link behavioral phenotypes to neural circuits, and would help to significantly improve the biological understanding of mouse locomotor kinematics.

While we agree that it would be quite powerful if such a rescue were possible, it would essentially amount to curing ataxia, which is clearly outside the scope of the current manuscript. Instead, what we propose is that applying this same approach to a broad range of mutants and circuit manipulations could achieve a similar goal. As we outline in the Discussion, we would argue that LD1 here represents “cerebellar dysfunction”, which is the shared anatomical feature in the two mutants, whereas LD2 would represent a combination of a) specific differences in the circuitry within the cerebellum itself, and b) extracerebellar effects. With the future addition of more mouse models and manipulations, the LDA would acquire more axes that could capture the contributions of other brain areas or cell types. We absolutely were not trying to imply that we had already achieved that broader goal, and we have tried to be more circumspect in our conclusions as suggested in major point #2. However, we emphasize that our approach has successfully captured and distilled the shared and specific locomotor phenotypes of two mutants with very clear, and previously well described, shared and specific neuropathologies – and to our knowledge this is the first time that this has been done. We hope to share the success of this approach with a wide audience in the hopes that others will build on the foundation we have established here – as they have already started to do with the LocoMouse system.

[Editors’ note: what follows is the authors’ response to the second round of review.]

Revisions for this paper:The reviewers found considerable merit in this advance but have further revisions they recommend that can be accomplished quickly and evaluated by the Reviewing Editor. Two of the reviewers thought that further experiments were needed to sustain the authors claims but realizing that this advance was considerable recommend revision in consultation.Reviewer 3:"The analyses provided reveal that some locomotor parameters are similarly impaired in these two mutant models while other parameters are unique to each mutant. While the additional findings presented in this study are sound, it is unclear how much we can generalize from analysis of one additional ataxia model, particularly because the neuropathology in these two models is quite different. Without a deeper evaluation of the generality and specificity of locomotor deficits in mouse models of ataxia, this study only provides a relatively slim additional advance over their previous study, and the more general conclusions they draw are not sufficiently justified. I have the following suggestions for strengthening the present manuscript. The authors show nicely in Figure 5 that combinations of behavioral features can be used to distinguish not only wild type from mutant mice but also pcd and reeler mice from each other. The authors' interpretation of these results is that the locomotor features that commonly distinguish wild type animals from the two mutants have a common underlying neuropathological origin (i.e. a cerebellar deficit) while the features that distinguish pcd mice from reeler mice reflect differences between these two mutants (i.e. absence of Purkinje cell output vs. miswiring in the cerebellum plus the additional extracerebellar defects in reeler mice). This is an important idea that could be bolstered with some additional data/analysis: 1. To show that aberrant cerebellar function is the root cause of the variability along LD1, the authors could use perturbation experiments to specifically affect locomotion by manipulating activity in interpositus-projecting Purkinje cells. This could be done acutely (as in Sarnaik and Raman, 2018) and/or chronically (as in Darmohray et al., 2018). 2. To test if LD1 does indeed distinguish wild type from ataxic mice and LD2 (or higher dimensions) distinguish different types of ataxic mice from each other, the authors could analyze locomotion in other models of ataxia that have different neuropathologies."Reviewer 2:"On a related note, a major claim is that the LD analysis defines a core set of behavioral features associated with cerebellar dysfunction. These are: impairments in multijoint, interlimb, and whole-body coordination, as well as in predictive mechanisms (i.e. forward models). It is true that these behavioral impairments are observed in both pcd and reeler mice, which share cerebellar neuropathology. However, it is possible that pathology in other brain circuits (basal ganglia, for example) may also be associated with some of the same behavioral impairments observed in the pcd and reeler mice, which would indicate that these behavioral impairments are not unique to the cerebellum after all. The impact of the paper, and the degree to which it is possible to claim that particular abnormalities are hallmarks of cerebellar dysfunction, would be much higher if the LD analysis could be repeated in mice with gait abnormalities caused by non-cerebellar dysfunction: Amende et al., 2005. Gait dynamics in mouse models of Parkinson's disease and Huntington's disease. If it is not feasible at this time to use the Locomouse system to analyze gait abnormalities in the mouse models in the Amende paper, the paper should at least discuss some of the limitations in terms of interpreting results as capturing behavioral hallmarks of cerebellar dysfunction. The distinction is between the impairments being "specific" markers of cerebellar dysfunction (which is supported by the data) vs "unique" markers of cerebellar dysfunction (which is not supported by the data). In other words, if a mouse has cerebellar dysfunction then we would be able to predict, based on the current paper, some of the abnormalities we may expect. On the other hand, if a mouse displays some of the abnormalities, it would not be possible to assume, based on the current results, that the underlying neuropathology is cerebellar." In light of these reservations, the authors should scale back some of their claims and discuss the limitations of the analysis more.

Thank you for these insightful comments and suggestions. We absolutely acknowledge that the conclusions that can be drawn from two mutants are limited and fully agree that the inclusion of additional cerebellar and non-cerebellar models within the framework we provide here will be hugely informative as to the specific contributions of both cerebellar and other circuits to locomotor phenotypes. Indeed, this has always been our plan, and we hope to follow up on this study with exactly those analyses in the near future. We have revised our language throughout the manuscript to soften our claims, address limitations, and make more explicit distinctions between what we have actually shown here, vs. predictions for future studies.